# Next-Generation Diagnostic with CRISPR/Cas: Beyond Nucleic Acid Detection

**DOI:** 10.3390/ijms23116052

**Published:** 2022-05-27

**Authors:** Pooja Bhardwaj, Rajni Kant, Sthita Pragnya Behera, Gaurav Raj Dwivedi, Rajeev Singh

**Affiliations:** ICMR-Regional Medical Research Centre, BRD Medical College Campus, Gorakhpur 273013, India; poojab288@gmail.com (P.B.); rajnikant.srivastava@gmail.com (R.K.); sp.behera1@gmail.com (S.P.B.)

**Keywords:** biosensor, diagnostics, CRISPR/Cas, nucleic acid, non-nucleic acid, endonuclease

## Abstract

The early management, diagnosis, and treatment of emerging and re-emerging infections and the rising burden of non-communicable diseases (NCDs) are necessary. The Clustered Regularly Interspaced Short Palindromic Repeats (CRISPR)-Cas system has recently acquired popularity as a diagnostic tool due to its ability to target specific genes. It uses Cas enzymes and a guide RNA (gRNA) to cleave target DNA or RNA. The discovery of collateral cleavage in CRISPR-Cas effectors such as Cas12a and Cas13a was intensively repurposed for the development of instrument-free, sensitive, precise and rapid point-of-care diagnostics. CRISPR/Cas demonstrated proficiency in detecting non-nucleic acid targets including protein, analyte, and hormones other than nucleic acid. CRISPR/Cas effectors can provide multiple detections simultaneously. The present review highlights the technical challenges of integrating CRISPR/Cas technology into the onsite assessment of clinical and other specimens, along with current improvements in CRISPR bio-sensing for nucleic acid and non-nucleic acid targets. It also highlights the current applications of CRISPR/Cas technologies.

## 1. Introduction

The recent example of the global COVID-19 outbreak provides insight into how pathogens are evolving into recalcitrant forms with enhanced pathogenicity [1]. Such situations necessitate prioritizing point-of-care diagnosis during outbreaks [2]. The real-time diagnosis of pathogens and non-communicable diseases (NCD) is an urgent need in clinical settings; however, due to inefficient diagnostic technologies, the diagnosis of pathogenic infections or NCDs at the initial stages is difficult, as well as at point of care (POC). This necessitates the development of POC diagnostic platforms. The World Health Organization (WHO) defined the ‘ASSURED’ (Affordable, Sensitive, Specific, User-friendly, Rapid and Robust, Equipment-free and Deliverable to end-users) criteria for the functioning of diagnostic products in poorly resourced regions where outbreaks are more likely to occur [3].

In recent years, numerous platforms for rapid nucleic acid (NA) detection have been developed; nevertheless, they may not be able to fulfil the ASSURED parameters at the same time [4]. Recently, many promising Clustered regularly interspaced short palindromic repeat-based diagnostics (CRISPR-Dx) (notably SHERLOCK, HOLMES, CASLFA, etc.) have been established which fulfils the WHO ASSURED criteria [5]. Though the majority of the literature is focused on the detection of NA by CRISPR-based systems, recent studies indicate that CRISPR-based systems also have the potential to diagnose using elements other than NA targets (proteins, antibiotics, monatomic ions and metabolites of amino acids, etc.) [2,6,7]. The present review was conceptualized to discuss the developments of CRISPR/Cas-based detection strategies utilizing different Cas subsystems, provide an overview, and investigate the mechanism of repurposing CRISPR/Cas for gene editing to the detection of biomolecules. This review extends the broad applicability of the CRISPR/Cas system to clinical as well as other key domains such as point-of-care tests (POCs).

## 2. CRISPR-Cas Endonuclease System: An Overview

The existence of CRISPR-associated nuclease (Cas) system was discovered in *Escherichia coli* by Japanese researchers in the late 1980s [5,8]. Subsequently, the CRISPR/Cas systems were also reported and recognized as acquired immunity systems from other bacterial strains and halophilic archaea [9,10,11]. These represent an unusual repetitive DNA sequence, and consequently were termed CRISPR by Jansen et al., in 2002 [12]. The recent classification of the CRISPR/Cas system is given in Figure 1 [13,14].

The most reliable and validated genome editing tools are type II, type V, and type VI CRISPR effectors [14,15,16,17]. Cas recognises and cleaves target sequences in DNA/RNA with the help of guide RNA (gRNA) [3]. Jennifer Doudna and Emmanuelle Charpentier (2020 Nobel Prize laureate) discovered CRISPR–Cas9 gene-editing tools in 2012 [18]. Their breakthrough redefined the CRISPR/Cas system for genome editing [18,19]. CRISPR–Cas9 has also been employed for genetic marker, typing, and epidemiological purposes, as an antimicrobial agent against pathogenic bacteria [16,19,20,21]. In addition to developing model cell lines and transgenic animals/plants, the CRISPR/Cas9 system’s flexibility allows it to be employed in understanding disease mechanisms, disease targets, and for transcriptional control [21]. Beyond these uses, Cas9 unique cleavage activity has also been exploited to construct ultrasensitive DNA biosensing platforms [3,22].

After 2014, many endonucleases were discovered, including Cas12b, Cas13a, CasRx, and Cas14 [16,23,24,25]. Single crRNA-guided cis (target nucleic acid) and trans (non-target nucleic acid) cleavage activity of Cas12a and Cas13a were discovered in the following years [26,27,28]. Recently, hypercompact CasΦ and Cas7-11 CRISPR/Cas systems were discovered. The CasΦ acts as a DNA cutter which was uniquely identified from genomes of huge bacteriophages and is half the weight of Cas9 and Cas12 [29], whereas Cas7-11 a new CRISPR Class 1 effector, originated from the fusion of Cas11 and Cas7 units derived from CRISPR subtype III-D. Similarly to Class 2 effectors, Cas7-11 processes crRNA and targets the RNA:Spacer duplex without any collateral activity [29,30]. A new perspective for using CRISPR-based diagnostics was promoted following the discovery of trans-cleavage activity (collateral cleavage activity) towards NA. The collateral cleavage of the ssRNA or ssDNA reporter results in a fluorescent signal as a readout, upon the activation of molecular sensors. By utilizing the collateral cleavage activity, several diagnostic platforms were developed including SHERLOCK, DETECTR, CARP, HOLMES [28,31,32]. Figure 2 summarizes the periodic discovery and development of diagnostic platforms.

## 3. Mechanism of CRISPR/Cas Based Detection

Classically, the CRISPR-Cas9 system is utilized by bacteria to destruct foreign DNA that enters the bacteria through horizontal gene transfer. Initially, a unique non-coding RNA trans-activating CRISPR RNA (tracrRNA) (transcribed separately) hybridizes to the precursor CRISPR transcript (pre-crRNA) through repeat sequences for post-transcriptional processing and endonucleolytic cleavage by RNaseIII to generate the CRISPR RNAs (crRNA) and forms a dual RNA hybrid structure [33]. This dual RNA guide directs Cas9 for dsDNA cleavage complementary to the target sequence located adjacent to PAM and cleaves each strand with distinct nucleases (HNH or RuvC) (Figure 3). Engineered sgRNA that combines crRNA and tracrRNA can simplify the system. The protospacer adjacent motif (PAM) sequence is NGG for CRISPR/Cas9, which occurs once in every 8 bps or 4 bps of random DNA, enables the design of sgRNA more conveniently [34]. An in vitro double-strand break (DSB) of target DNA caused by the CRISPR/Cas9 foreign DNA destruction programme was a breakthrough and created the basis for its gene-editing toolbox application [35]. This characteristic of Cas9 activation upon target recognition via gRNA possesses great potential in genotyping and gene detection. The ‘magic power’ of Cas9 in biosensing applications has already been established [34,36,37,38].

Cas13 nucleases target ssRNA and are a Type VI CRISPR-Cas signature protein. The exclusion of the tracrRNA requirement is the distinctive feature of Cas13 effectors (HEPN)-binding domains (Figure 4A). Cas13a has been shown to possess dual RNAse activity. First RNAse activity leads to crRNA maturation, while second RNAse activity is responsible for RNA-activated single-stranded RNA destruction [39,40]. The presence of dual RNase activity of Cas13 is due to the presence of two RNase catalytic pockets, namely the REC lobe (containing 1-helical domain) which cleaves pre-crRNA and the NUC lobe (containing two HEPN domain) that recognize and cleave target RNA [40,41]. The mechanism of cleavage (target (cis) and non-target RNA (trans) lies in the recognition and binding of Cas13 to a protospacer flanking site (PFS) adjacent to the complementary spacer on the 5’ and 3’ ends of the protospacer for Cas13b nuclease, whereas for Cas13a it is present on the 3’ end of ssRNA. The Cas13 endonuclease enzyme with crRNA induces blunt-end breakage and degrades arbitrarily non-target ssRNA present nearby [42]. A recent report demonstrated the dependence of cleavage activity on conserved catalytic residues in the two HEPN domains as well as the preference of uracil residue for cleavage [39].

Cas12, the type V CRISPR-Cas protein, has a single RNA-guided RuvC domain. The Cas12 effectors are a close relative of Cas9, with a single RuvC nuclease domain, enabling Cas12 to introduce staggered dsDNA breaks [25]. Cas12a and Cas12b of the Cas12 family are widely studied. Cas12b, but not Cas12a, requires tracrRNA for catalysis, whereas nuclease activity in both of them is activated by a PAM sequence upstream of its target. After crRNA-target binding to protospacers juxtaposed to the PAM consensus, Cas12 RuvC-like nucleases cleave the non-target strand within the protospacer 18 nucleotide (nt) from the PAM using a two-metal ion (Figure 4B) [43]. The crRNA/Cas12 complex cleaves dsDNA in a staggered fashion, leaving 5- to 7- nt overhangs [44,45,46].

Cas14 (Cas12f) is a type V CRISPR/Cas system that evolved from a superphylum of symbiotic archaea and was discovered to be a distant relative of Cas12 [16,47]. Cas14 targets ssDNA with trans ssDNA cleavage. Harrington et al., discovered that Cas14 evolved independently as the smallest protein (400–700 amino acids) in comparison to other Cas endonucleases (950–1400 amino acids) and that it clusters into the following three subgroups: Cas14a (Cas12f1), Cas14b (Cas12f2), and Cas14c (Cas12f3) [16,48]. Cas14 was also called a miniature CRISPR system due to its size. Cas14 is a single effector molecule and has been found to destruct DNA independently without requiring PAM (Figure 4C) [16,42,49]. Karvelis et al., on the other hand, demonstrated that a T-rich PAM sequence can induce the Cas14 endonuclease to cleave dsDNA with staggered ends [48,50]. Although Cas14 possesses conserved functional RuvC nuclease, it lacks the RNAse III enzyme, which processes crRNAs and tracrRNAs, as validated by a biochemical study [16].

Likewise, the newly discovered bacteriophage-derived CasΦ endonuclease is another hypercompact (70 to 80 kDa) endonuclease with a single RuvC active site for both pre-crRNA processing and DNA cleavage. CasΦ is bigger than Cas14 (40- to 70-kDa) but half the size of other CRISPR-Cas systems (100- to 200-kDa). Unlike Cas14, CasΦ requires T-rich PAM sequences (5’-TBN-3’). For the activation of CasΦ endonuclease activity, a cognate PAM and a 14 nt spacer are required. Similar to type V CRISPR-Cas endonucleases, CasΦ generates staggered 5’-overhangs of 8 to 12 nt. Furthermore, when activated by cis recognition, CasΦ efficiently showed non-specific cleavage activity towards dsDNA and ssDNA within the RuvC active site [29].

## 4. CRISPR/Cas Tools for Nucleic Acid Detection

Beyond gene editing, CRISPR-based systems have been extensively explored for NA detection. The next section of the review provides details about the NA detection tools that have been developed with different CRISPR/Cas systems (Table 1).

### 4.1. CRISPR/Cas9

The CRISPR/Cas9 system has been explored for the detection of NA (Figure 3). The potential of Cas9 in DNA detection was demonstrated using the Cas9-based reverse PCR (CARP) technique [38]. Following which, the CRISPR/Cas9-based PCR-dependent DNA detection method, including CRISPR-typing PCR (ctPCR) and the CRISPR/Cas9-triggered isothermal exponential amplification reaction (CAS-EXPAR) were developed [36,50,51]. The developed platforms were sensitive and specific enough but were restricted to laboratory use due to difficulties in developing reverse PCR primers, bulky apparatus usage, time consumption, or the use of RNAse sensitive enzymes. If modified, this technology can be used for point-of-care testing (POCT). The CRISPR/Cas9-mediated lateral flow nucleic acid assay (CASLFA) turned out to be time-efficient (1 h) and sensitive (100 gene copies). The CRISPR/Cas9 system combined with recombinase polymerase amplification (RPA) for target gene amplification and lateral flow detection allowed for the visual recognition of the target at an ambient temperature [34].

### 4.2. Deactivated CRISPR/Cas9

dCas9 (dead or dCas9) was first created by mutating the nucleases domain HNH and RuvC of *S. pyogenes* Cas9 [33,35]. With abolished nuclease activity, it still possesses the ability to detect target DNA with the help of sgRNA (Figure 3). This activity has been successfully implemented for many CRISPR/Cas-based biosensor platforms [22,54,55].

A pair of dCas9 proteins was utilized to develop an in vitro method for *Mycobacterium tuberculosis* DNA detection based on the luminescence signal [55]. They linked the reporter N- and C-terminal halves of the firefly luciferase enzyme (NFluc and CFluc) with two dCas9, directed by engineered sgRNA, complementary to the upstream and downstream segments of a target DNA sequence. When target DNA is bound to paired dCas9 on the DNA segment within proximity (20 bp space), a luminescence signal is generated due to luciferase catalytic activity [54]. A Methicillin-resistant *Staphylococcus aureus* (MRSA) detection technique was developed based on the dCas9/sgRNA-SYBR Green (SG) I-based DNA-FISH system [55]. This technique is simple and can directly detect MRSA from cell lysate without the need for DNA extraction, thus saving a lot of valuable time but with a limitation of usage in terms of the heavy instruments required for fluorescence intensity measurements [55]. A more improved, facile, rapid (20 min), and specific biosensor for scrub typhus (ST) and severe fever with thrombocytopenia syndrome (SFTS) was developed with a sensitivity of 0.54 aM and SFTS of 0.63 aM, respectively [22]. This biosensing measures changes in the refractive index (RI). The target-specific primer was immobilized onto the surface of a silicon microring resonator (SMR) biosensor, and isothermal amplification with RPA was performed in a reaction chamber containing dCas9 Ribo Nucleoproteins (RNPs). The duplex (CRISPR/dCas9:target template) increased RI with each resonant wavelength after hybridization with an immobilized primer. This provides 100-times more sensitivity than RT-PCR in detecting the target pathogen [22]. In 2019, Hajian et al., developed a method for amplification-free NA detection using CRISPR–Chip [56].

### 4.3. CRISPR/Cas12a

The collateral cleavage activity of Cas12a effector proteins has been harnessed for the development of quick, sensitive, specific and reliable NA-detection platforms such as HOLMES (One-Hour Low-cost Multipurpose highly Efficient System), and DETECTR (DNA Endonuclease-Targeted CRISPR Trans Reporter). Both platforms uses the *Lachnospiraceae bacterium* Cas12a protein (LbCas12a) [27,28,58]. Studies suggested PAM (TTTA) sequence is a pre-requisite for a crRNA-complementary dsDNA catalytic activation but not a crRNA-complementary ssDNA. Cas12a also exhibited better cleavage selectivity with a shorter crRNA guide sequence (20 nucleotides) [27,58,78]. Based on CRISPR-Cas12a:RPA, Ai et al., devised a diagnostic approach to identify MTB infection with a detection limit of around a single-copy of genes (five copies µL^−1^) [59].

Cas12a collateral activity was pivotal in efforts to simplify the diagnostic approach. Without compromising sensitivity or specificity, CRISPR-Cas12a provided a visual interpretation of the results via a turbidity, blue light, or lateral flow analysis (LFA) (Figure 5) [32,60,79]. To prevent aerosol contamination during the uncapping of the reaction tube, the visual Cas12a-based platform “Cas12a-VDet” was devised. New biosensing technologies facilitated optical detection with DETECTR-like sensitivity. Pathogen detection and visual interpretation at isothermal temperatures make CRISPR/Cas platforms appropriate for POCT [60]. Bai et al., presented CORDS (Cas12a based On-site and Rapid Detection System) for ASFV detection with high specificity and a detection limit of 10 aM [32]. Following Bai et al.’s lead, another research team used filter-paper-based DNA extraction to develop LFA with RPA-LbCas12a for pathogen diagnosis and GMO administration in rice [61].

### 4.4. CRISPR/Cas12b

Cas12b, another subtype of the CRISPR V subsystem, was used for the development of a more advanced version of HOLMES, i.e., HOLMESv2.0. HOLMESv2.0 incorporates Loop-Mediated Isothermal Amplification (LAMP) for the isothermal amplification of target and the Cas12b protein, derived from *Alicyclobacillus acidoterrestris* (AacCas12b), for facile NA detection. It is a one-pot, one-step method. Cas12b has a distinct target preference compared to Cas12a and a stronger ssDNA trans-cleavage activity (10 min) than dsDNA [20]. AapCas12b, discovered in *Alicyclobacillus acidiphilus*, exhibits excellent catalytic activity and selectivity for dsDNA. The AaCas12b-based detection platform (CDetection), based on the AaCas12b-sgRNA-dsDNA-activator system, was used to efficiently detect dsDNA. Teng et al., reported HOLMESv2.0 as being less sensitive than the CDetection platform, while CDetection was superior to DETECTR in terms of sensitivity. CDetection achieved single-nucleotide sensitivity in DNA detection by employing tuned gRNA (tgRNA) [71]. A thermophilic Cas12 effector is necessary because the LAMP reaction proceeds isothermally but at a high temperature (60–68 °C). AacCas12b and AapCas12b both demonstrated dsDNA cleavage at high temperatures, hence, they are good choices for aid with LAMP PCR [80].

### 4.5. CRISPR/Cas13

The Cas13a-integrated collateral effect and isothermal amplification laid the foundation for the SHERLOCK platform (Figure 4A) [26]. Zika and Dengue virus detection, pathogenic bacteria, human DNA genotype, and the identification of mutations in cell-free tumor DNA were all successfully demonstrated by Gootenberg et al., using SHERLOCK [26]. SHERLOCK has an attomolar sensitivity and single-base mismatch specificity. Cas13a’s collateral cleavage (reporter nucleic acid) has long-term and multiple turnovers, thus facilitating signal amplification even in the presence of a single gene copy and subsequently leading to ultra-sensitivity [81]. The HUDSON platform has been linked with SHERLOCK to facilitate the direct and instrument-free detection of Zika and Dengue viruses from clinical samples [82]. SHERLOCKv2.0 was devised as a follow-up study by Gootenberg et al., This variant has a sensitivity of 2 aM and produces lateral strip readouts for $0.61 per paper-based test [73,83].

SHERLOCK prepared the foundation for many upcoming NA-detection platforms such as SHERLOCKv2.0, CARVER and CARMEN [84]. SHERLOCKv2, CARVER and CARMEN Cas13 have been found to possess inherent power for multiplexing [73,74,76]. Generally, the CRISPR-Cas platform uses fluorescence as a readout, hence a low-cost, portable instrument such as a portable fluorescence detector is necessary. To solve this issue, Katzmeier and colleagues created a pocket-sized fluorescence reader costing less than $15 [84]. 

### 4.6. CRISPR/Cas14

Harrington et al., created the Cas14-DETECTR device to demonstrate high-fidelity SNP genotyping [16]. Cas14 substrate identification requires a complementary seed sequence. There is much to explore about the implication of Cas14 in diagnostics. Cas14 may be an efficient CRISPR/Cas system for identifying and detecting distinct ssDNA viruses infecting hosts in all three domains of life or transposable elements proliferating via ssDNA intermediates through a multiplex approach [16,85,86].

## 5. CRISPR/Cas for Non-Nucleic Acid Targets Detection

Beyond gene editing and nucleic acid detection, Cas12a has also shown its proficiency for non-nucleic acid targets (NNTs) detection [2,87]. The robust mechanism of CRISPR/Cas12a towards signal magnification makes it a promising tool for the detection of NNTs such as biomolecules, proteins, exosomes and ions (Figure 6, Table 2). Signal amplification with CRISPR/Cas depends on NA-target recognition, hence the detection of NNTs relies indirectly on the biological reaction of activating the trans-cleavage activity of Cas effectors, probe affinity and sensor presentation. Although, the detection of NNT with CRISPR biosensors remains a challenge due to matrix interference, low LOD, and sensitivity; and to overcome these challenges, biosensing platforms for NNT have been developed using CRISPR technology.

### 5.1. Aptamer Based Detection of NNTs

Aptamers are short oligonucleotide sequences (20–60) capable of binding to a wide range of compounds, ranging from simple inorganic molecules to large protein complexes [97]. CRISPR/Cas-based NNTs detection aided with aptamers has been widely recognized. For instance, the Cas12a endonuclease-based aptamer approach has been utilized for Adenosine triphosphate (ATP) detection [87,91]. ATP serves as the main source of energy and can be used as an indicator for hygiene, cell viability, and food quality control and therefore in biochemical and medical applications [98,99]. In this approach, generally, the target DNA for the Cas endonuclease is locked with the aptamer specific for NNTs. In the presence of the target molecule, aptamers bind to the molecule, leaving the Cas target unlocked, hence activating the CRISPR/Cas system for cleaving the labelled fluorophore reporter, resulting in a fluorescent signal for detection. Likewise, Peng et al., demonstrated the detection of ATP and Na+ with the CRISPR biosensing platform aided with Cas12a:crRNA:ssDNA substrate (labelled with a fluorophore and quencher at each end (ssDNA-FQ); functional (fDNA)). In the absence of target molecules, fDNA remains bound by DNA activator and keeps Cas12a activity locked, while the presence of NNT triggers the unlocking of the DNA activator, thereby activating the collateral cleavage of ssDNA-FQ by Cas12a. This results in an enhanced fluorescent signal which can be read by portable fluorometers available on the market [91].

The alpha femtoprotein (AFP) was recognised as a clinical biomarker for hepatocellular carcinoma, and germ cell tumors were utilized for cancer diagnosis using Cas12a with aptamers for analyte detection [89]. Exosomes are types of extracellular vesicles of exosomes-secreting cells, typically composed of DNA, RNA and proteins and are closely related to cancers [100]. Based on exosome marker protein CD63, Zhao et al., developed a CRISPR/Cas12a-based detection platform for exosome detection using CD63 aptamers. The aptamers here were blocked with a blocker specific to sgRNA for Cas12 activation. When the exosome was recognised, the aptamer connected to the exosome binding protein, releasing the blocker, which was then detected by CRISPR/Cas12, initiating its trans-cleavage activity and generating a fluorescent signal for confirmation [89].

### 5.2. aTFs Based NNTs Detection

Bacterial allosteric transcription factors (aTFs) are sensitive effectors that can sense and respond to small molecular entities. The biosensing of small molecules such as uric acid and p-hydroxybenzoic has been achieved using CaT-SMelor. The aTFs here were fused and mobilized onto cellulose. The dsDNA-containing PAM motif as well as the binding domain corresponding to aTFs designated as functional DNA (fDNA), remain bound to aTFs blocking the CRISPR activation. The presence of NNT brings about conformational changes to aTF and the release of fDNA, leading to the activation of CRISPR DNase which releases the signal for detection. CaT-SMelor demonstrated the detection of uric acid (causative molecule of Gout) directly in clinical human blood specimens [2].

Drugs such as cocaine (benzoylmethylecgonine) are extensively used in drug abuse, representing another global problem. To monitor illegal drug supplies and consumption, and to detect the presence of drugs in suspected samples, a CRISPR/Cas12a-based platform was presented by Zhao’s group [89]. A new approach relying on SHERLOCK-based profiling (Cas13a) for small molecule detection method was developed named “SPRINT” [7]. Remarkably, SPRINT, can detect eight different analytes which include cofactor, antibiotic (tetracycline), nucleotides, monatomic ion and metabolites of amino acid in clinical samples. SPRINT utilizes the principle of aTF and riboswitches to block the transcription of RNA polymerase. The analyte binding to aTF or riboswitch permits conformational changes and frees RNA polymerase to proceed to transcription. Transcript (mRNA) generated acts as an activator Cas13a endonuclease, inducing the collateral cleavage of labelled reporter ssRNA [7].

### 5.3. Enzyme Substrate-Based NNTs Detection

Barcoding the NA with an enzyme-substrate offers the prompt detection of marker enzymes in clinical samples. Hao et al., demonstrated the in-vivo detection of human colon adenocarcinoma with the help of urine as a biomarker. Here, the DNA was barcoded with the protease substrate for proteases specifically synthesized by cancerous cells. With the help of transcriptomics, CRC-specific proteases—exclusively expressed in tumor-bearing cells—were identified and accordingly the NA was chemically modified with protease substrate. Upon substrate cleavage, DNA molecules concentrate in the urine sample, which when subjected to CRISPR/Cas12 detection generate fluorescent as well as strip-based signals confirming the cancerous stage [88].

Another important checkpoint that controls various biological processes such as DNA damage repair is the phosphorylation/dephosphorylation of biomolecules. Polynucleotide kinase/phosphatase (PNKP) is an important repair enzyme regulating DNA replication and repair, and its irregular expression may lead to diseases such as cancer [95]. Considering PNKP as a target for cancer recognition (HeLa cells), Wang et al., designed substrate dsDNA with 3’end modification with the phosphate group. The presence of PNKP only enables the phosphorylation of the 3’end followed by the extension of DNA, whereas the absence of PNKP does not trigger DNA extension, which further activates Cas12a/crRNA trans-cleavage activity resulting in fluorescent signal amplification for detection. They demonstrated the system’s ability to detect PNKP activity at the single-cell level [95]. Various markers for cancerous cells can thus be utilized in future to develop a POCT for precise and accurate early cancer-detection platforms.

### 5.4. Peptides Based NNTs Detection

Recently, the CRISPR/Cas system was harnessed for in vitro protein interaction studies (Figure 7A–C). dCas9 fused Peptide libraries are barcoded with unique sgRNA, multiplexed for viral epitope mapping, and can be customized for rapid large scale protein studies. This platform, which was referred to as PICASSO, can locate user-programmed sites on a microarray surface containing DNA sequences complementary to each peptide’s sgRNA from a single mixed pool of peptides fused with dCas9 [6]. The design of the assays includes the co-expression of dCas9 fusion peptides and uniquely designed sgRNA on the same plasmid so that each *E. coli* produces a pair of sgRNA and dCas9 fusion peptides, which can be isolated from a single mixed culture of *E.coli*. The dCas9:peptide:sgRNA locates the complementary dsDNA (pre-assembled on the microarray platform) then binds irreversibly. Hence, a large number of complex peptide libraries can be self-assembled on a microarray chip and the screening of complex peptides can occur at the same time. After applying a dCas9-sgRNA library to a dsDNA microarray, PICASSO permits the screening of mutagenesis peptide libraries in a couple of hours with accuracy, eliminating the requirement of Next-generation sequencing [6].

### 5.5. ELISA Based NNTs Detection

With the emergence of ideas for diagnosis, Chen et al., developed a CRISPR/Cas13a-based ELISA platform; CLISA demonstrated the detection of human IL-6 and human vascular endothelial growth factor (human VEGF); an important factor involved in disease development. In CLISA, the Antibody-Antigen-Antibody complex formation takes place as in sandwich ELISA (Figure 7D). The enzyme (such as horseradish peroxidase) here is replaced with a biotinylated dsDNA bearing a T7 promoter sequence. The sequential addition of streptavidin and biotinylated dsDNA to the complex results in the binding of the DNA amplification template to the complex. The addition of T7 RNA polymerase then leads to the amplification of the DNA template to the RNA substrate liable for Cas13a/crRNA-mediated fluorometric detection. CLISA has demonstrated the detection of human IL-6 and human VEGF detection at femtomolar concentration.

Although the CLISA is sensitive, rapid and specific compared to ELISA, caution has to be taken to provide an RNAse free environment as the reporter/template used for detection is RNA, which is more prone to degradation with RNase contamination [101].

## 6. Broad Application of CRISPR/Cas System

Several breakthroughs in diagnostic strategies have been implemented over time. NA sequence analysis led to accurate illness diagnosis and treatment [102]. The CRISPR/Cas endonuclease system has emerged as a strong pathogen diagnosis tool. As a result of this detection method, bio-sensing platforms for pollutants in water/food, mutation detection, genotyping, plant genes detection, antibiotic resistance and microbiome alteration have been developed (Figure 8). The most remarkable feature of CRISPR/Cas is its ability to discriminate between the single base substitution and its recognition of many sequences at once with extreme precision. To date, CRISPR/Cas transformed the diagnostics from lab settings to public resource systems following the ASSURED parameter.

### 6.1. Diagnosing Viral Infections with CRISPR/Cas System

An explosion in the number of reports utilizing CRISPR/Cas12 Platforms for the development of COVID-19 detection kits occurred during the COVID-19 pandemic. The First indigenous kit from India, the FnCas9 Editor Linked Uniform Detection Assay (FELUDA) based on CRISPR/Cas9, was developed for COVID detection, which in association with a mobile app, can be readily interpreted at home itself [53]. Besides Cas9, several CRISPR/Cas13 and Cas12-based COVID-19 detection platform such as STOPCovid, CREST, SHINE, SHERLOCK and DETECTR for COVID, VanGuard, AIOD, DISCoVER and many more have been developed for the same purpose [1,62,63,64,65,75,77,103,104]. Various CRISPR/Cas-based platforms aided with isothermal amplification have been utilized for the detection of Porcine reproductive and respiratory syndrome virus (PRRSV) [67], ASFV [32,105], Zika/Dengue virus [73], HIV [65], HPV16/18 [66,96] BK polyomavirus and Cytomegalovirus [106].

Taking CRISPR to the next level, multiplexing can be introduced for the simultaneous detection of multiple targets. Despite its insurmountable design, researchers believe CRISPR/Cas multiplex biosensing is a feasible goal [107]. SHERLOCK V2.0 introduced CRISPR multiplexing for Zika/Dengue virus ssRNA, genotyping, and mutation detection in patient biopsies [73]. It may take a while to establish such a process because it requires a range of Cas effectors, each with a unique cutting profile. CARMEN, the most powerful multiplexed CRISPR/Cas13, was developed and demonstrated simultaneous differentiation between 169 human-associated viruses, Influenza A strains subtyping and the multiplexed identification of dozens of HIV drug-resistance mutations [76]. A new variant of Cas13, named miniature Cas13 (mCas13) (837 aa), was identified using a metagenome analysis. The mCas13 effectors were synthesized and cloned to produce proteins synthetically. This compact effector molecule was tested for SARS-CoV-2 diagnosis and exhibited comparable results to CRISPR/Cas13 systems [108]. Furthermore, this scenario of rapid response to pandemics highlights how demand pushes researchers to build platforms that are urgently required in times of crisis. The wax-printing technology, AuNP-labelled (both end) probe for Smartphone app-based result readouts demonstrated its potential for next-generation-diagnostics (NGD) [109].

### 6.2. Diagnosing Bacterial Pathogen and Bacterial Drug Resistance

With dCas9, the first approach for TB detection was initiated following which the CRISPR/Cas12 effector was utilized as a pulmonary and extra-pulmonary TB-detection platform [9,59,68,110,111]. CRISPR-based detection eliminates pathogen cultures and provides faster results without requiring an expert technician. The detection of other WHO listed pathogens such as MRSA and MSSA via the CRISPR/Cas system has also been investigated [56]. Other diseases which are considered neglected but important to track in terms of pathogens include ST, and STFS. For this purpose, researchers have taken the initiative to focus their research on developing a CRISPR-based detection assay [22].

### 6.3. CRISPR/Cas for Parasite Detection

Parasites play a crucial role in vector-borne diseases. Hence, the tracking of vector-borne pathogens is necessary for controlling infections. To keep an eye on such issues, the development of a SHERLOCK-based assay for diagnosing the malarial parasite has been attempted. The assay was able to detect pan Plasmodium, *P. falciparum* and *P. vivax* by considering *18S rRNA* as the target and successfully demonstrated its applicability via clinical samples [112]. However, the approach used by Cunningham et al., failed in one pot detection. To overcome this, another research group developed a field-deployable one-pot method for the detection of *P. falciparum, P. vivax, P. ovale, and P. malariae*, out of which *P. falciparum* and *P. vivax* showed 100% specificity via lateral flow readout/handheld spectrophotometer [113]. To diagnose the parasite that infects pacific white leg shrimp, the RPA-CRISPR/Cas12a fluorescent reporter system targeting *ptp2* gene of EHP was developed [114].

### 6.4. CRISPR/Cas for Non-Communicable Disease Diagnosis

Cancer is a multifaceted disease resulting from cellular genetic alteration and a heterogeneous microenvironment. Early cancer detection can ensure proper treatment and decreased mortality. The available techniques for cancer detection lag behind in terms of time, sensitivity and specificity. But the Cas endonuclease system with its multiplexing capability, sensitivity and specificity can facilitate multiplexed mutation detection in very little time [50]. microRNAs (miRNA) play an important part in regulating various biological routes via post-transcriptional gene expression and can therefore be utilized as biomarkers for cancer diagnosis [115]. Using Extracellular vesicles derived miRNA as targets and Cas9 endonucleases as a detection system, a diagnostic platform with RCA-assisted CRISPR/Cas9 cleavage (RACE) for the screening, diagnosis, and prognosis of cancer cell detection was developed [37]. In concordance with this, another RCA-based biosensing platform assisted with CRISPR/Cas12a cleavage was developed for exosomal miRNA detection for the prognosis of breast cancer. The system demonstrated sufficient performance (34.7 fM) in laboratory and clinical samples, but lacks a real-time detection system and hence cannot be applied to the POC diagnostics [116].

Peng and researchers conducted their studies with miRNA-21 as a model target. With two DNA hairpins, they devised a CHA (catalytic hairpin assembly) circuit to facilitate its coupling with CRISPR/Cas12a. The presence of target miRNA can lead to the formation of an H1/H2 duplex containing the PAM sequence 5′-TTTA-3′, which further activates Cas12a/gRNA endonuclease collateral cleavage activity, thereby generating fluorescence signals. This devised CRISPR-CHA biosensing system demonstrated its capability in diagnosing different cancer cells and clinical serum samples with sub-femtomolar sensitivity [93]. Similarly, miR-17 as a target and CRISPR/Cas13a were utilized to develop the hyper-branching rolling circle amplified CRISPR/Cas13a biosensor (HyperCas) platform. HyperCas showed 500-fold superior detection sensitivity, an ultralow limit-of-detection (LOD) of 200 aM, and an accurate detection of miR-17 from cancer cell extracts with the potential to discriminate between homologous miRNA-17 family members [94]. Further, the has-miRNA-31, which is associated with oral squamous cell carcinoma (OSCC), is utilized as a target for the CRISPR detection strategy for OSCC detection termed as “ISAR/Cas12a-dmStrip” [95]. The hairpin DNA (hDNA) complementary to target miR31 was designed in such a way that in the presence of target miR31 only, the hDNA open up and with the help of the invading stacking (IS) primer, the target was amplified generating multiple copies of dsDNA using a single miRNA only. The amplified target was then utilized as a substrate by Cas12 endonuclease, thereby activating trans-cleavage and generating a signal for confirmation [94]. 

Based on CRISPR-chip, Uygun et al., demonstrated the detection of circulating tumor DNA using dCas9/sgRNA immobilized on Grapheneoxide screen-printed electrodes (GPHOXE) targeting the tumor-related PIK3CA exon 9 mutation which was analyzed by electrochemical impedance spectroscopy [56,57]. Modern CRISPR technology does not rely only on the nucleic acid targets but non-nucleic acid targets have also been used for diagnosing cancer, as described in earlier sections [88,95,116].

### 6.5. CRISPR-Cas for Agriculture

Plant genome editing using CRISPR/Cas9 has been widely demonstrated. However, plant pathogen diagnostics is still trailing behind. The early diagnosis of plant pathogens may help farmers control plant disease and reduce crop losses. Chang et al., using CRISPR/Cas, demonstrated the detection of *Phytophthora infestans*, which is a water mould pathogen known for causing infections in tomatoes or potatoes via colorimetric detection [51]. CRISPR Multiplexing has been not only limited to clinical pathogens but broadened its scope to plants too. The classification of the Soyabean variety has been demonstrated using the SHERLOCK strategy with two different Cas13 effectors, for two different genes, namely the CP4 *EPSPS* gene and *LE1* gene, in a single reaction [117]. Likewise, Zhang et al., demonstrated plant pathogen detection and Bt-transgenicity identification in GMO Rice with the Cas12a system [61].

Both RNA and DNA viruses infecting plants have been targeted using CRISPR/Cas12a effectors. Diagnostic platforms for the detection of DNA viruses including *tomato yellow leaf curl virus* (TYLCV) and *tomato leaf curl New Delhi virus* (ToLCNDV), RNA viruses such as *Apple necrotic mosaic virus* (ApNMV), *Apple stem pitting virus* (ASPV), *Apple stem grooving virus* (ASGV), *Apple chlorotic leaf spot virus* (ACLSV), potexvirus, potyvirus, and tobamovirus and viroids such as *Apple scar skin viroid* (ASSVd) have been developed with the aid of the CRISPR/Cas12a effector [118,119,120]. A procedure of co-infection with multiple viruses in an apple tree with minimal sample preparation was demonstrated [119]. The development of such noble, ideal, cost-effective, rapid and reliable detection strategies that can eliminate the requirement of a specialized laboratory is of utmost importance to decrease the loss in crop productivity.

### 6.6. CRISPR/Cas for SNP Genotyping and Mutation Detection

The screening of mutations is time and money consuming. It is feasible to identify the biallelic mutant accurately in a cost-effective manner using the Cas12a-based biosensing platform with a sensitivity of single-base resolution [121]. The CRISPR/Cas9 system can be utilized for target sequencing as realized by Quan and colleagues, who developed FLASH (Finding Low Abundance Sequences by Hybridization)-NGS. Using FLASH, antimicrobial resistance genes were targeted and covered in a single sequencing [52]. With the capacity of Cas endonuclease to discriminate single nucleotide and collateral cleavage activity of Cas12 (HOLMESv2.0) and Cas14 (Cas14-DETECTR), the CRISPR system has been repurposed for SNP’s genotyping and detection [16,20].

### 6.7. CRISPR for Food Adulteration

Food adulteration applies to any food item that puts public health at risk. The concern of food adulteration is being studied from various perspectives such as food quality and safety, food fraud and food defence [122]. The key players of diagnostic settings can also be repurposed for identifying adulterated food.

*Listeria monocytogenes*, which is one of the most virulent foodborne pathogens, has been detected successfully with precision via CAS-EXPAR and CASLFA detection systems [34,50]. In addition to pathogen detection, CAS-EXPAR showed remarkable excellency in differentiating single base-mismatch and DNA methylation detection [50]. Shen et al., developed the platform ‘APC-Cas’ for *Salmonella enteritidis* detection in milk and mouse serum. They designed the allosteric probe (AP) containing the following three domains: the Aptamer domain for facilitating target recognition, Primer binding site and T7 promoter domain. The AP remains in the bound dimension until the target pathogen was present. In the presence of the target, AP unwinds and releases the primer binding site, which on further polymerization and Cas endonuclease action generates a strong fluorescence signal [123]. Subsequently, Peng’s group demonstrated the detection of *Streptococcus aureus* in milk with Cas12a endonuclease [124]. However, again, the necessity of the pre-amplification of the target through PCR limits its usage to laboratory settings.

## 7. Advancements in CRISPR Detection

According to the studies, the pre-amplification of nucleic acid is required to obtain a stronger signal and avoid erroneous results [20,28,71]. Moreover, a recent study suggested that using pooled crRNA to target multiple regions of the target could obviate the need for pre-amplification [43]. Furthermore, the CRISPR/Cas12 dynamics influence not only NA and biomolecule detection but also protein detection with Cas12a. The isothermal proximity CRISPR Cas12a assay is a new CRISPR technique that recognises the target (protein or NA) based on proximity binding rather than crRNA recognition (iPCCA). Upon binding, it allows for primer extension, which results in the generation of a barcode (a pre-designed CRISPR targetable sequence) for signal amplification [69]. The proximity binding approach makes it possible to detect NNTs without PAM sequences. With specifically designed primers that meet the design requirements of LAMP or RPA, PAM sequences can be delivered to the target during pre-amplification. Any target (with or without the PAM sequence) can theoretically be detected using PAM-introduced primers for pre-amplification [68,125]. Furthermore, cross-contaminations can be avoided with visual or smartphone-based applications for result readouts [53,109].

## 8. Future Perspective

Bacterial defence mechanisms are being repurposed. CRISPR/Cas systems are now capable of detecting more than genomic material. The CRISPR/Cas detection platform can be used to examine non-genomic targets such as enzymes, proteins and analytes. There are several sectors where the CRISPR/Cas system can be used that have yet to be investigated. Contamination detection in food/water samples, plant pathogen detection, microbiome monitoring, rare mutations in human samples, emerging antibiotic pathogen detection, cancer mutations, rare mosaic allele detection, targeted transcriptomics from clinical samples, and the recovery of targeted transcripts from single-cell sequencing libraries are just a few of the untapped research areas where CRISPR/Cas effectors may prove useful [52,117].

All recent studies established the CRISPR/Cas system’s potential and demonstrated that it may be repurposed for the detection of contaminants, enzymes, proteins, analytes, and plant diseases, rather than being limited to clinical settings and nucleic acids. Several platforms have yet to be translated into clinical applications and must be brought into clinical settings from the laboratory. These CRISPR/Cas biosensing platform results output may be connected to unique cloud data for storage and processing purposes via 5G services of mobile phone apps to assist societies in healthcare situations. CRISPR/Cas detection technologies, when combined with an artificial intelligence (AI)-driven model, could facilitate the early detection and identification of dangerous diseases, as well as generate an alarm to alert health experts [102,126].

## 9. Conclusions

Only a few CRISPR/Cas systems have been uncovered so far, but the microbial world has thrived because of this built-in defence machinery. Understanding microorganisms’ defence machinery and its implications for diagnostics requires a dedicated investigation in this emerging field. The success of such research hinges on a thorough understanding and effective application of the CRISPR/Cas system.

## Figures and Tables

**Figure 1 ijms-23-06052-f001:**
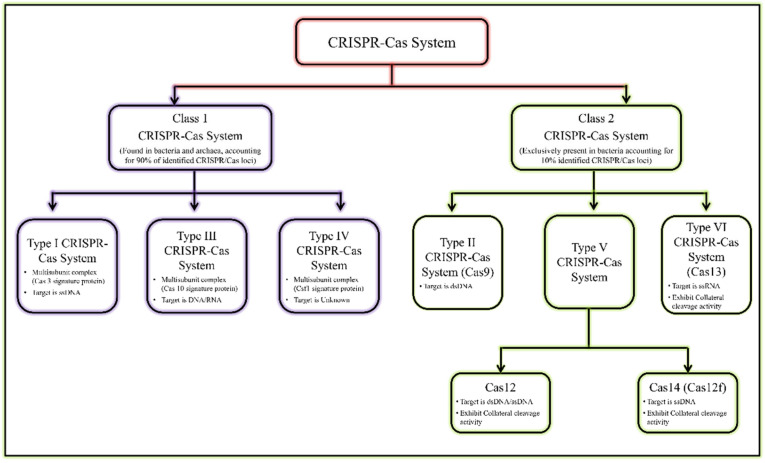
General classification of CRISPR/Cas system.

**Figure 2 ijms-23-06052-f002:**
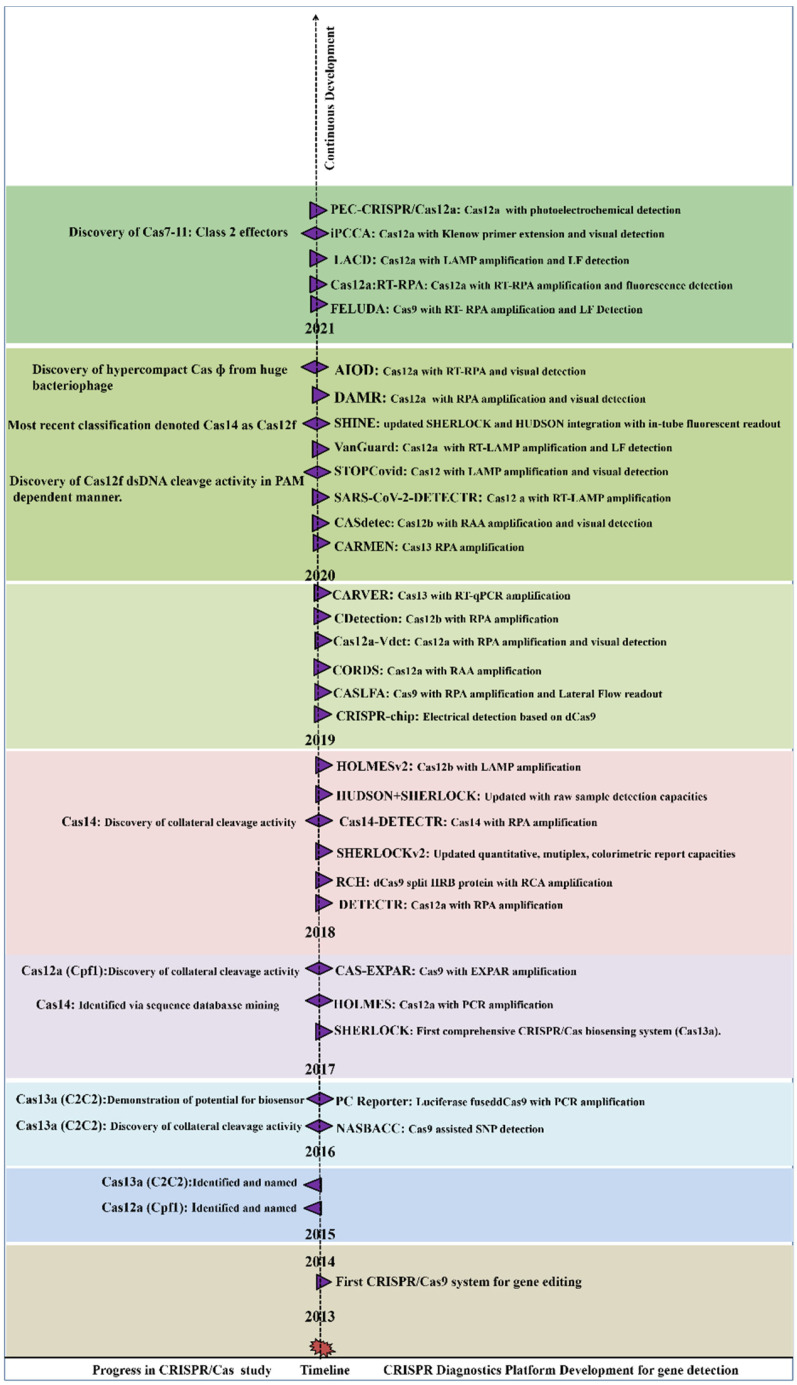
Year-wise Discovery and development of CRISPR/Cas based Nucleic acid detection platform.

**Figure 3 ijms-23-06052-f003:**
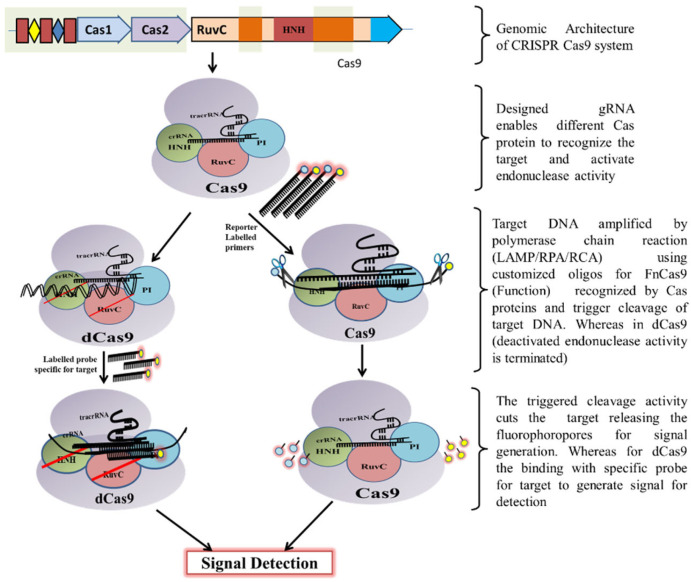
Schematic diagram showing CRISPR/Cas9 mechanism for DNA detection.

**Figure 4 ijms-23-06052-f004:**
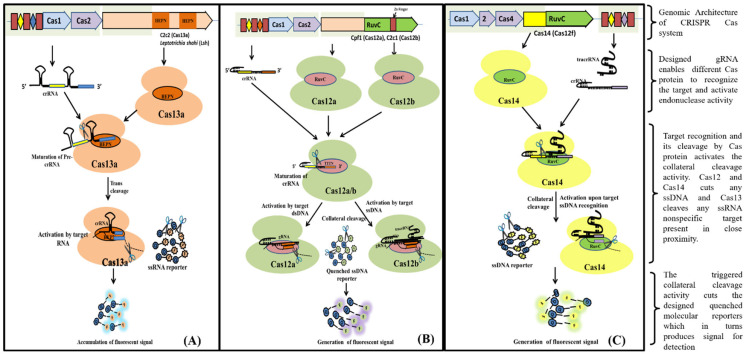
Schematic diagram showing various steps from Cas endonuclease formation to activation of its collateral cleavage activity. (**A**) Mechanism of CRISPR/Cas13 cis-activation upon target RNA detection activating it for collateral-cleavage of non-target ssRNA. (**B**) Mechanism of cis-activation of CRISPR/Cas12a and CRISPR/Cas12b by target dsDNA and ssDNA respectively. Both CRISPR/Cas12a and Cas12b trans-cleave non-target ssDNA upon cis-activation. (**C**) Mechanism of CRISPR/Cas14 (Cas12f) cis-cleavage activation by target ssDNA detection and trans-cleavage activation for non-target ssDNA upon cis-activation.

**Figure 5 ijms-23-06052-f005:**
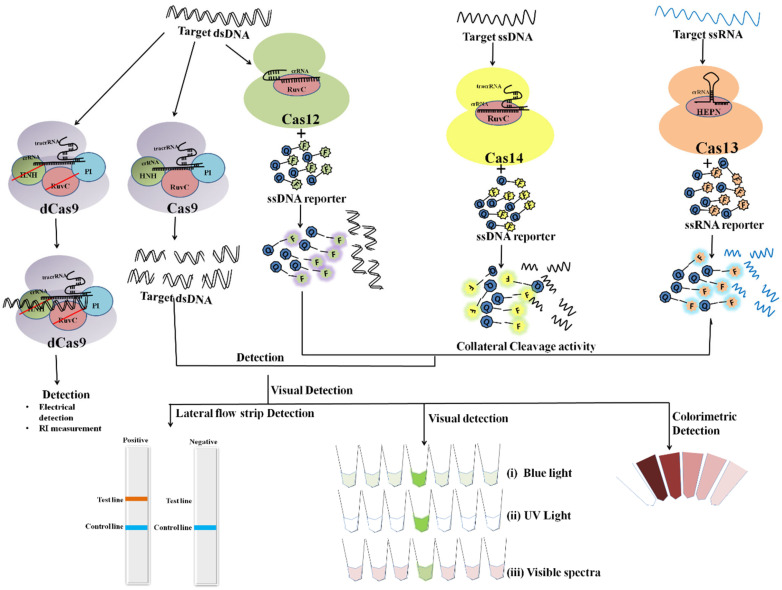
Overview of different CRISPR/Cas effector mechanisms with various result interpretation strategies.

**Figure 6 ijms-23-06052-f006:**
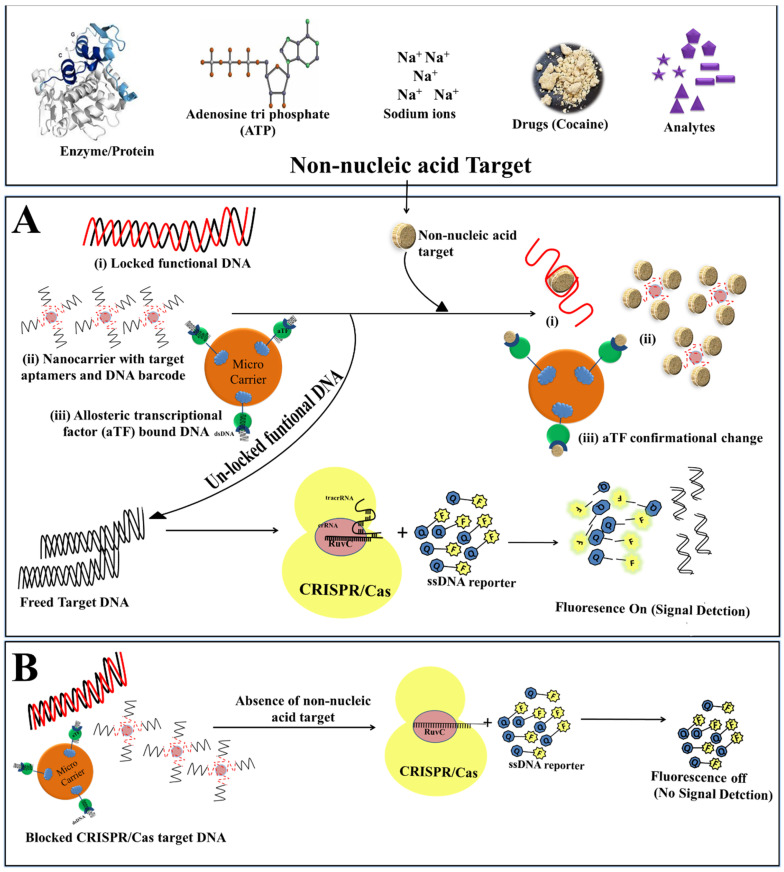
CRISPR/Cas based bio-sensing mechanism for non-nucleic acid targets (NNTs). (**A**) Bio-sensing mechanism of CRISPR/Cas system in the presence of NNTs. The binding of NNTs to allosteric transcriptional factors brings about conformational changes and releases the locked target DNA, which activates CRISPR/Cas endonuclease. This activates the collateral cleavage activity of the CRISPR/Cas system and chop-off the reporter DNA to produce the signal. (**B**) The absence of target NNTs does not trigger the release of target DNA hence no signal generates.

**Figure 7 ijms-23-06052-f007:**
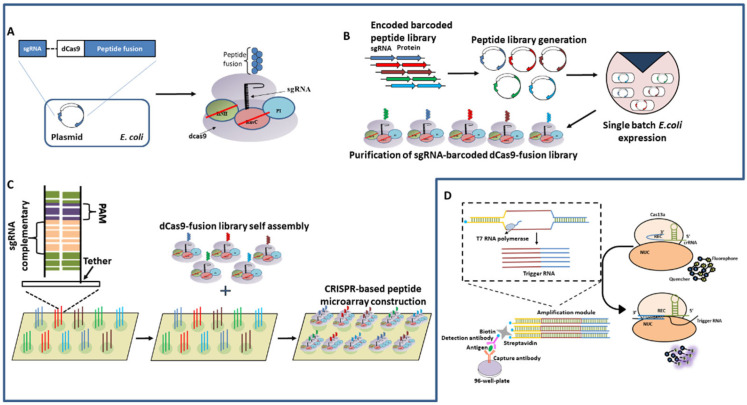
(**A**–**C**) CRISPR-based peptide display and microarray self-assembly by PICASSO; (**A**) A peptide fused to dCas9 is barcoded with a unique sgRNA that is expressed in *E. coli*. (**B**) Overview of sgRNA-barcoded dCas9-fusion peptide library synthesis. Peptide-encoding sequences are barcoded with unique sgRNAs, introduced into an expression vector in a single mixed pool, and synthesized and purified from a single batch of *E. coli.* (**C**) sgRNA-barcoded dCas9-fusion peptides in a mixture self-assemble on a corresponding DNA microarray surface, enabling quantitative protein studies using customized peptide collections. (**D**) CLISA Platform.

**Figure 8 ijms-23-06052-f008:**
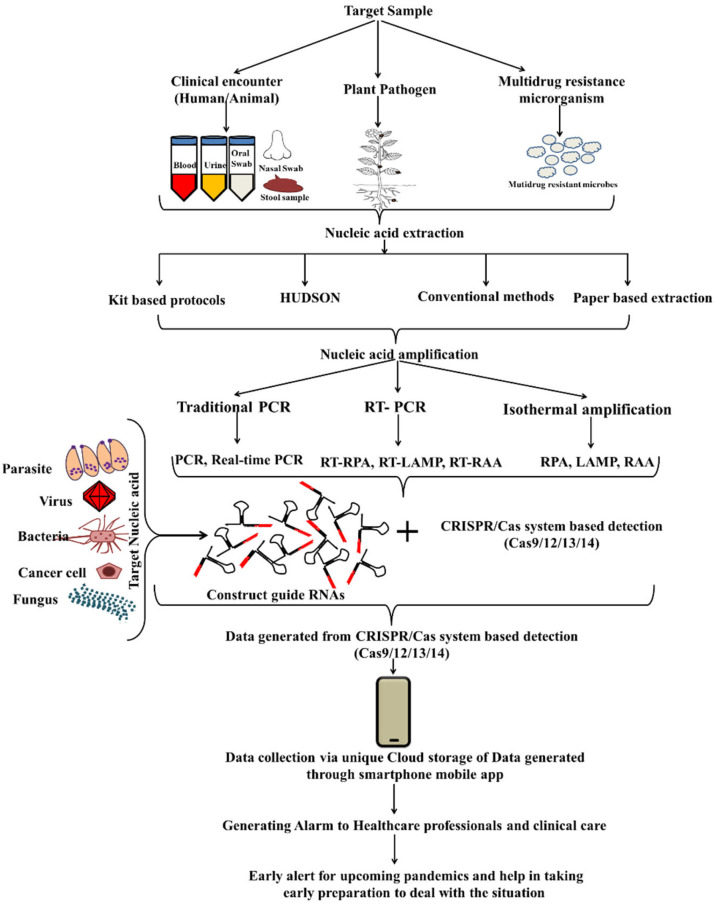
Flow-chart presenting the various optional steps required for CRISPR/Cas based detection and its future application.

**Table 1 ijms-23-06052-t001:** List of CRISPR/Cas based nucleic acid detection platforms developed so far for diagnostics purposes.

S. No.	CRISPR-Cas Platform	Cas Enzyme	Sensitivity	Nucleic Acid Amplification	Detection	Result Output	Reference
1	CAS-EXPAR	Cas9	0.82 amol	Real-time PCR	DNA methylation and *Listeria monocytogenes (**L. monocytogenes)*	Real time fluorescence monitoring	[50]
2	CtPCR		400 copies/μL	PCR	Human Papillomavirus (HPV)	SYBR Chemistry or Gel electrophoresis	[36]
3	CRISPR/Cas9 for plant pathogen		2 pM	SDA, RCA	*Phytophthora infestans*	Colorimetric	[51]
4	FLASH		aM	NGS	AMR	Sequence read	[52]
5	RACE		fM	RCA	EV- miRNAs	Fluorescence	[37]
6	CASLFA		<100 copies/μL	RPA	African swine fever virus (ASFV), genetically modified organisms (GMO), and *L. monocytogenes*	LFA	[34]
7	FELUDA		-	RT-RPA	SARS-CoV2	LFA	[53]
8	CRISPR/dCas9	dCas9	5 × 10^−^^5^ nmol/mL	PCR	*Mycobacterium tuberculosis*	luminescence	[54]
9	dCas9/sgRNA- Sybr Green (SG) I		10 cfu/mL	NR	Methicillin-resistant *Staphylococcus aureus*	Fluorescence	[55]
10	dCas9/sgRNA-SMR		ST (0.54 aM) and SFTS (0.63 aM);	RPA/RT-RPA	Scrub typhus and Severe fever with thrombocytopenia syndrome	Refractive index change	[22]
11	CRISPR–Chip		1.7 fM	NR	Duchene muscular dystrophy	Electrical detecion	[56]
12	GPHOXE		0.65 nM	-	Circulating tumor DNA	Electrochemical impedance spectroscopy	[57]
12	HOLMES	Cas12a	1–10 aM	PCR	DNA virus pseudorabies virus (PRV) and RNA viruses (Japanese encephalitis virus (JEV), SNP	Fluoresence	[28]
13	DETECTR		N/A	RPA	HPV	Fluoresence	[58]
14	CRISPR-Cas12a:RPA		5 copies/μL	RPA	*Mycobacterium tuberculosis*	Fluoresence	[59]
15	Cas12a-VDet		10 aM	RPA	mycoplasma	Visual	[60]
16	CORDS		10 aM	RAA	ASFV	LFA	[32]
17	CRISPR/Cas12a		1 pg	RPA	GMO rice	LFA	[61]
18	STOPCOVID		~2 aM	LAMP	SARS-CoV2	LFA, Fluoresence	[62]
19	SARS-CoV-2 DETECTR		90%	RT-LAMP	SARS-CoV-2	Fluoresence detection	[63]
20	VaNGuard (Variant Nucleotide Guard)		-	RT-LAMP	SARS-CoV-2 with mutation detection	LFA	[1]
21	CRISPR/Cas12a		10 copies/μL	rRT-PCR	SARS-CoV-2	Fluoresence	[64]
22	AIOD		1.2 copies	RT-RPA, RPA	SARS-CoV-2 and HIV	Visual	[65]
23	DAMR		10–100 Copies	RPA	HPV16/18	Visual/Fluoresence	[66]
24	Cas12a:RT-RPA		1 copy	RT-RPA	PRRSV	Fluoresence	[67]
25	LACD		-	LAMP	*Mycobacterium tuberculosis*	LFA, Fluoresence	[68]
26	iPCCA		1 fM	Nicking Cleavage & primer extension by Klenow fragment	Interleukin-6 gene	Fluoresence	[69]
27	PEC-CRISPR/Cas12a		0.4 fM	-	HIV	Photo electrochemical	[70]
28	HOLMESv2.0	Cas12b	-	LAMP	SNP	Fluoresence	[20]
29	CDetection		0.1–1 aM	RPA	HPV	Fluoresence	[71]
30	CASdetec		1 × 10^4^ copies/mL	RAA	SARS-CoV-2	Visual, Fluoresence	[72]
31	SHERLOCK	Cas13	~2 aM	RPA, RT-RPA	Zika and Dengue virus, genotype human DNA, and mutations in cell-free tumor DNA	Fluoresence detection	[26]
32	SHERLOCKv2.0	Cas13/Cas12/Csm6	~2 aM	RPA	Zika and Dengue virus	LFA	[73]
33	CARVER	Cas13	-	RT-qPCR	lymphocytic choriomeningitis virus; influenza A virus; and vesicular stomatitis virus	Fluoresence	[74]
34	SHINE		90%	RT-RPA	SARS-CoV-2	In-tube Fluoresence, LFA	[75]
35	CARMEN		Attomolar	RPA	Human associated RNA viruses	Fluorescence microscopy	[76]
36	CREST		-	PCR	SARS-CoV-2		[77]
37	Cas14-DETECTR	Cas14	-	phosphorothioate (PT)–containing primer amplification	SNP genotyping	Fluoresence detection	[16]

The ‘-’ in the table refers to non-availaibility of the information.

**Table 2 ijms-23-06052-t002:** List of CRISPR/Cas based non-nucleic acid detection platforms developed for diagnostics purposes.

S. No.	CRISPR-Cas Enzyme (Platform)	Sensitivity	Non-Nucleic Acid Target (Aided with)	Application	Result Output	References
1	Cas12a	-	Urine biomarker (proteases)	Cancer	Lateral flow	[88]
2	Cas12a	of 0.21 μM (ATP) and 0.10 mM (Na+)	ATP and Na+	Hyponatremia. Hypernatremia	Fluorescent	[87]
3	Cas12a	0.24–977 fM	Alpha femtoprotein (AFP) was utilized (aptamers)	hepatocellular carcinoma and germ cell tumors	Fluorescent	[89]
4	Cas12a	3 × 10^3^–6 × 10^7^ particles per microliter	Exosome membrane protein (CD63 aptamer)	Lung cancer	Fluorescent	[90]
5	Cas13a (SPRINT)	-	Nucleotides, metabolites of amino acids, tetracycline and monatomic ions (using allosteric transcription factors (aTF) and riboswitches)	Transcriptional activator function	Fluorescent	[7]
6	Cas12	400 nM	ATP (aptamer)	Biochemical and medical applications.	Fluorescent	[91]
7	Cas12a (CaT-SMelor)	10 nM	uric acid and p-hydroxybenzoic acid (aTFs)	Gout	Fluorescent	[2]
8	Cas12a (iPCCA)	100 fM	interleukin-6 protein	Allergy	Fluoresence	[69]
9	Cas12a (CRISPR-CHA)	0.07 fM	miRNA-21	Cancer	Fluorescence	[92]
10	Cas13a (HyperCas)	200 aM	miRNA-17	Cancer	Fluorescence	[93]
11	Cas9 (RACE)	34.7 fM	Vesicle miRNA	Cancer	Fluorescence	[37]
12	Cas12a (ISAR/Cas12a-dmStrip)	aM	Salivary has -miR31	oral squamous cell carcinoma	Fluorescence, LFA	[94]
13	Cas12	3.3 × 10^−6^ U/mL	Polynucleotide kinase/Phosphatase	DNA damage repair-related biological enzyme	Fluorescence	[95]
14	Cas12 (3D DNA walker)	0.331 fM	miRNA-141	Cancer	Electrical	[96]

The ‘-’ in the table refers to non-availaibility of the information.

## Data Availability

Not applicable.

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
