# Peer review of "Next-Generation Diagnostic with CRISPR/Cas: Beyond Nucleic Acid Detection"

_ijms, 2022, doi:10.3390/ijms23116052_

Round 1

Reviewer 1 Report

The manuscript with manuscript ID: ijms-1725262 entitled " Next generation diagnostic with CRISPR/Cas: Beyond nucleic acid detection" seems good review. This review is well written and has sufficient information and may be published in the International Journal of Molecular Sciences after minor revision because it deals with the novel aspects of use of CRISPR/Cas system (non-nucleic acid biosensing).

The article well discusses non-nucleic acid (miRNA, protein, small molecule, analyte, and hormone) detection platform, which is a new area of research. However, non-nucleic acid biosensing is in the stage of infancy, but this review will justify the readers about this forthcoming area of research to plan future tactics. According to me, this article summarizes the following

  • Provide insight into CRISPR/Cas Mechanism
  • Provides an overview of the various applications of CRISPR/Cas technology to date (till 2021).
  • Throws light on the future area of research such as non-nucleic acid detection.
  • Explains how CRISPR/Cas is a suitable candidate for the development of the point of care test as well as to handle the pandemic situation
  • Describes the newly discovered Cas endonucleases

Further suggestions;

  • Because writers have not documented the detection techniques for CRISPR/Cas bio-sensing separately anywhere in the review, so authors are advised to provide a separate overview of the detection strategies utilised for CRISPR/Cas to inform readers about the strategies available and what can be expected.
  • The mechanism section solely mentions Cas9, Cas13, and Cas12a & b, with no mention of other CRISPR/Cas mechanisms such as Cas14f. The authors are advised to include the mechanism of CRISPR/Cas14 and newly discovered Cas endonucleases (Cas7-11 and Casɸ).

I have some points for the authors to consider:

  1. However, this paper is self-explanatory and can be acceptable with revision after typographical and grammatical corrections.
  2. The manuscript is on the whole well written but there are some problems with the English (including tenses, plurals, matching of adjectives and nouns, adverbs and verbs) such that other sections are almost ‘good’.
  3. In introduction in Line no. 26, there is repetition of words for eg; Recent was used twice in a single line kindly rewrite.
  4. Additional space problem, Kindly check throughout the manuscript. Few instances are:

                        Line 60: discovered CRISPR–Cas9

Line 139: CRISPR/Cas9 system has been explored

Line 169: sensitivity of 0.54

Line 170: (22). In 2019 Hajian

Line 183: LbCas12a) (27,28,54). Studies

Line 242: Cas12 (70). Cas14

Line 248: 16,42,72). Karvelis et al., on the

Line 280: ATP serves as the main

Line 424: printing technology, AuNP labelled

  1. Spacing is required between

Line 296: Exosomes;a type

Line 414: take awhile to

Line 514: base resolution(117).

Line 517: single sequencing(118).

Line 522: at risk.The

Kindly check and correct throughout the manuscript

  1. Missing preposition:

Line 394: CRISPR/Cas is their ability discriminate the

  1. Missing punctuation

Line 404: Besides Cas9 several CRISPR

  1. Kindly cite the correct reference in Line 408: ASFV 32,98, Zika.
  2. Line 270: Caption for Figure 6 should be more descriptive (Panel A & B not described individually in caption).
  3. Line 432, 559: CRISPR/cas should be written as CRISPR/Cas
  4. If the authors make the suggested changes, then accept this article because being as important contribution on CRISPR/Cas. If they do not then the article will be yet another of those ‘tantalising’ article that promises much but deliver little and therefore do not get cited.

Author Response

Reviewer 1

The manuscript with manuscript ID: ijms-1725262 entitled " Next generation diagnostic with CRISPR/Cas: Beyond nucleic acid detection" seems good review. This review is well written and has sufficient information and may be published in the International Journal of Molecular Sciences after minor revision because it deals with the novel aspects of use of CRISPR/Cas system (non-nucleic acid biosensing). The article well discusses non-nucleic acid (miRNA, protein, small molecule, analyte, and hormone) detection platform, which is a new area of research. However, non-nucleic acid biosensing is in the stage of infancy, but this review will justify the readers about this forthcoming area of research to plan future tactics. According to me, this article summarizes the following

  • Provide insight into CRISPR/Cas Mechanism
  • Provides an overview of the various applications of CRISPR/Cas technology to date (till 2021).
  • Throws light on the future area of research such as non-nucleic acid detection.
  • Explains how CRISPR/Cas is a suitable candidate for the development of the point of care test as well as to handle the pandemic situation
  • Describes the newly discovered Cas endonucleases

Authors Reply: Authors are very thankful for the learnt reviewer for the the valuable revision and critical comments to improve the quality of the review.

Comment 1: Because writers have not documented the detection techniques for CRISPR/Cas bio-sensing separately anywhere in the review, so authors are advised to provide a separate overview of the detection strategies utilised for CRISPR/Cas to inform readers about the strategies available and what can be expected.

Authors Reply: Thanks for the suggestion. The principle behind the detection of CRISPR/Cas activity for interpretation (visual/colorimetric/lateral flow) is its collateral cleavage (trans-cleavage) activity which has been explained and discussed in the various section of the manuscript for example:

Section 2: Line no 80-83: “A new perspective for using CRISPR-based diagnostics was opened up by the discovery of trans-cleavage activity (collateral cleavage activity) towards NA. The collateral cleavage of ssRNA or ssDNA reporter results in a fluorescent signal as a readout, upon activation of molecular sensors”

Section 4.2: Line no. 210-213 “Cas12a collateral activity played an important role in efforts to simplify the diagnostic approach. Without compromising sensitivity or specificity, CRISPR-Cas12a provided visual interpretation of the result via turbidity, blue light, or lateral flow analysis (LFA)”

In view of the above CRISPR/Cas activity was described many places, due to this addition of another section is not crucial in this MS.

Comment 2: The mechanism section solely mentions Cas9, Cas13, and Cas12a & b, with no mention of other CRISPR/Cas mechanisms such as Cas14f. The authors are advised to include the mechanism of CRISPR/Cas14 and newly discovered Cas endonucleases (Cas7-11 and Casɸ).

Authors Reply: As per learnt reviewer suggestion the mechanism of Cas14 & Casɸ has now been incorporated in the revised manuscript. Whereas the newly discovered CRISPR/Cas system (Cas7-11) is reported for gene-editing and is free of collateral cleavage activity. The newly discovered Cas endonucleases yet have not been repurposed for the development of a detection platform, hence the authors did not include the mechanism of the newly discovered Cas7-11 endonucleases.

Reference:

Özcan A, Krajeski R, Ioannidi E, Lee B, Gardner A, Makarova KS, et al. Programmable RNA targeting with the single-protein CRISPR effector Cas7-11. Nature [Internet]. 2021 Sep 6 [cited 2021 Sep 23]; Available from: https://www.nature.com/articles/s41586-021-03886-5

Comment 3: I have some points for the authors to consider:

  1. However, this paper is self-explanatory and can be acceptable with revision after typographical and grammatical corrections.
  2. The manuscript is on the whole well written but there are some problems with the English (including tenses, plurals, matching of adjectives and nouns, adverbs and verbs) such that other sections are almost ‘good’.
  3. In introduction in Line no. 26, there is repetition of words for eg; Recent was used twice in a single line kindly rewrite.
  4. Additional space problem, Kindly check throughout the manuscript.

Authors Reply: Thanks for excellent suggestion, now the grammatical errors in the revised MS have been checked and corrected throughout the manuscript at our best.

Reviewer 2 Report

The review article by Pooja Bhardwaj et al. titled “Next generation diagnostic with CRISP/Cas: Beyond Nucleic acid detection,” describes, in great detail,  Clustered Regularly Interspaced Short Palindromic Repeats (CRISPR)-Cas systems and its various applications to the microbial and clinical world.  There are various types of CRISPR-Cas system classifications, but they all have an underlying utilization: A genome editing tool that uses Cas enzymes and guide RNA, gRNA, to cleave target DNA or RNA. The review properly depicts the general classification of CRISPR/Cas system in the opining Figure. Without CRISPR-Cas technology, rapid nucleic acid (NA) detection would be challenging at the initial stages at point of care (POC) and diagnosing pathogens and non-communicable diseases (NCD) would be inefficient due to unsatisfactory diagnostic products available in poorly resourced areas.

Also, the paper brought awareness to CRISPR/Cas system potential, outside the clinical setting; for instance, when combined with artificial intelligence, one could detect more dangerous diseases. The conclusion was strongly written and was consisted with all argument presented. Article stressed how CRISPR systems are impactful, and yet demonstrated there needs to be more investigation regarding this topic. The future of this technology capabilities are promising, and its efficiency will substantially increase, leaving a greater global impact.

In my opinion, this review, article highlighting the purpose of the various CRISPR systems, will strongly contribute to the scientific community and  enhance interest to the IJMS. This work fits well to the journal's scope. The manuscript is clear, organized, and well written and therefore, I recommend this work for publication.

Author Response

Authors are very thankful to the learnt reviewer for the valuable and kind words.

Round 2

Reviewer 1 Report

The revised version of the manuscript includes all remarks and modifications indicated. The main concerns of the manuscript have been solved. In my opinion, the provided version is now suitable for publication